# CT abnormalities 3 and 12 months after hospitalization for COVID-19 and association with disease severity: A prospective cohort study

Trond Mogens Aaløkken[1,2], Haseem Ashraf[2,3], Gunnar Einvik[2,4], Tøri Vigeland Lerum[2,5], Carin Meltzer[6], Jezabel Rivero Rodriguez[6], Ole Henning Skjønsberg[2,5], Knut Stavem [2,4,7]*

1 Department of Radiology and Nuclear Medicine, Oslo University Hospital Rikshospitalet, Oslo, Norway, 2 Institute of Clinical Medicine, University of Oslo, Oslo, Norway, 3 Department of Diagnostic Imaging, Akershus University Hospital, Lørenskog, Norway, 4 Pulmonary Department, Akershus University Hospital, Lørenskog, Norway, 5 Department of Pulmonary Medicine, Oslo University Hospital Ullevål, Oslo, Norway, 6 Department of Radiology and Nuclear Medicine, Oslo University Hospital Ullevål, Oslo, Norway, 7 Health Services Research Unit, Akershus University Hospital, Lørenskog, Norway

* knut.stavem@medisin.uio.no

**Data Availability Statement:** Data availability: The ethical approval granted by the Regional Committees for Medical and Health Research

## Abstract

### Objectives

To investigate changes in chest CT between 3 and 12 months and associations with disease severity in patients hospitalized for COVID-19 during the first wave in 2020.

### Materials and methods

Longitudinal cohort study of patients hospitalized for COVID-19 in 2020. Chest CT was performed 3 and 12 months after admission. CT images were evaluated using a CT severity score (CSS) (0–12 scale) and recoded to an abbreviated version (0–3 scale). We analyzed determinants of the abbreviated CSS with multivariable mixed effects ordinal regression.

### Results

242 patients completed CT at 3 months, and 124 (mean age 62.3±13.3, 78 men) also at 12 months. Between 3 and 12 months (n = 124) CSS (0–12 scale) for ground-glass opacities (GGO) decreased from median 3 ($25^{th}$–$75^{th}$ percentile: 0–12) at 3 months to 0.5 (0–12) at 12 months (p<0.001), but increased for parenchymal bands (p<0.001). In multivariable analysis of GGO, the odds ratio for more severe abbreviated CSS (0–3 scale) at 12 months was 0.11 (95%CI 0.11 0.05 to 0.21, p<0.001) compared to 3 months, for WHO severity category 5–7 (high-flow oxygen/non-invasive ventilation/ventilator) versus 3 (non-oxygen use) 37.16 (1.18 to 43.47, p = 0.032), and for age $\geq$60 compared to <60 years 4.8 (1.33 to 17.6, p = 0.016). Mosaicism was reduced at 12 compared to 3 months, OR 0.33 (95%CI 0.16 to 0.66, p = 0.002).

Ethics in Norway does not allow public sharing of the data. A data set can be made available for scientific analysis on request, provided data is handled in accordance with ethical regulations (written ethics protocol, full compliance with the Declaration of Helsinki). To ensure full anonymity only the main variables of the final analyses are provided. We confirm that the data file provided constitutes the minimal data set necessary to replicate the findings of our study in their entirety. Data requests can be made to corresponding author Knut Stavem (knut.stavem@medisin.uio. no) or data custodian, Haldor Husby (haldor. husby@ahus.no).

**Funding:** This study had financial support from Boehringer-Ingelheim, Norway. The funders had no role in study design, data collection and analysis, decision to publish, or preparation of the manuscript.

**Competing interests:** Haseem Ashraf reports grant from Boehringer-Ingelheim, Norway. Knut Stavem reports personal fees from UCB Pharma and MSD Norway, unrelated to this study.

## Conclusions

GGO and mosaicism decreased, while parenchymal bands increased from 3 to 12 months. Persistent GGO were associated with initial COVID-19 severity and age $\geq$60 years.

## Introduction

Computed tomography (CT) provides detailed information about the extent and pattern of lung involvement in COVID-19. In the acute phase, CT scans often reveal ground-glass opacities (GGO), consolidations, and other signs of inflammation [1–3]. There is a considerable concern for sequelae in survivors, particularly reduced lung function and structural changes, such as chronic pulmonary fibrosis [4,5].

Recent reviews and meta-analyses of CT findings 6–12 months after COVID- have reported a wide range of prevalence rates for post-COVID-19 CT abnormalities [6–8]. After 1 year, the prevalence of chest CT lung sequelae ranges from 7–97%, with GGO ranging from 2–68% [8]. Fibrotic-like findings are also common and show little change in prevalence 4–7-months to 1 year after COVID-19 [7]. Most studies with 1-year follow-up have reported the prevalence or change in overall or specific CT abnormalities [9–13]. Some studies also investigated determinants of persistent CT abnormalities, such as fibrosis, in multivariable logistic regression models [9,12,14–17].

In interpreting CT images, many studies used systematic scoring systems to describe the severity and distribution of abnormalities [18,19]. These scales are typically qualitative scales with ordinal properties, but they are often dichotomized in further analysis. This approach fails to utilize the full spectrum of available data, thereby diminishing the statistical power of the analyses [20,21]. The striking heterogeneity in findings across studies [7,8,22], to a large extent due to selection bias and publication bias [8], and the underuse of available information, underscores the need for more studies to address the temporal changes in chest CT and the association with disease severity in COVID-19 survivors.

We have previously presented a cohort of patients hospitalized for COVID-19 in 2020, including follow-up with data on dyspnea, lung function, and a crude analysis of CT abnormalities [17]. The present study aimed to determine the chest CT severity scores and temporal changes for pulmonary CT abnormalities between 3 and 12 months, and to investigate their association with disease severity in multivariable analysis in this cohort.

## Material and methods

### Study design, population and procedures

All study participants provided informed consent through a paper form or via a secure web-application. The study was approved by the Regional Ethics Committee for South-Eastern Norway (no. 2020/125384) and the Data Protection Officer at each participating hospital. It is registered with ClinicalTrials.gov (NCT 04535154).

This longitudinal observational cohort study included patients $\geq$18 years of age recruited from six Norwegian hospitals during the first wave of COVID-19 as part of the "Patient-Reported Outcomes and Lung Function After Hospitalization for COVID-19" study (PRO-LUN) [17,23]. Recruitment was consecutive for patients discharged from hospital from 11 March 2020 to 1 June 2020, by obtaining consents 4–8 weeks after discharge. We excluded

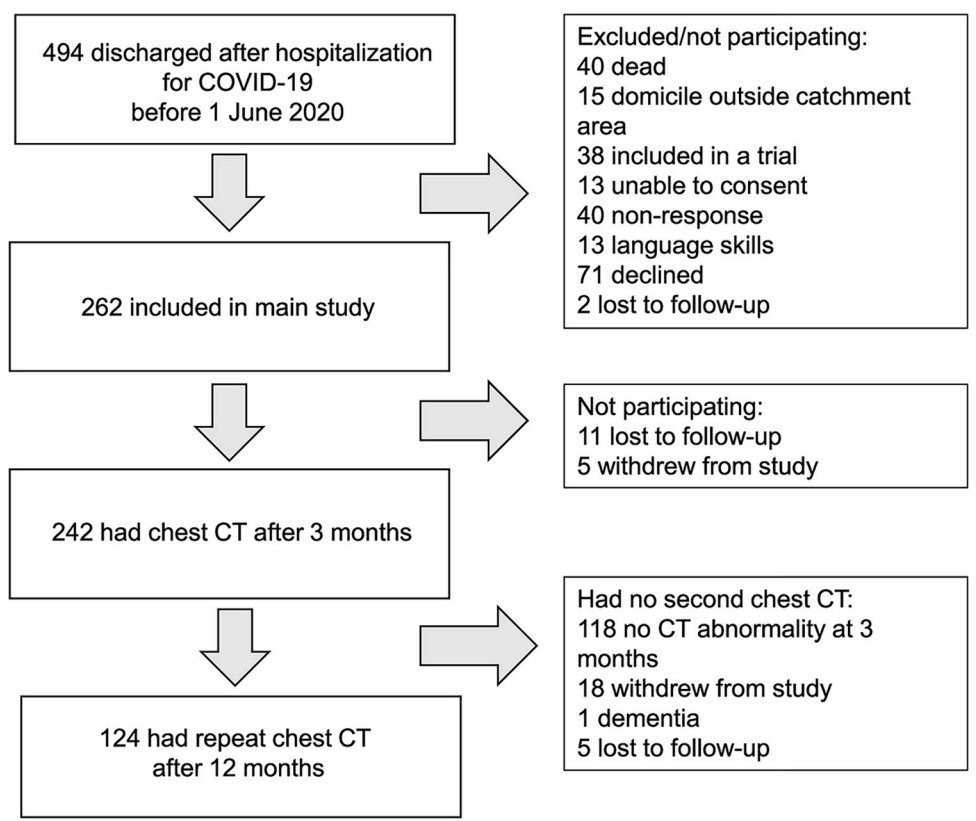

**Fig 1. Flow-chart of patient inclusion and attrition in the study.**

patients that had domicile outside the hospitals' catchment areas or participated in a clinical trial.

Baseline variables during the acute COVID-19 were collected by review of the electronic medical records. Comorbidity was scored using the Charlson comorbidity index [24]. Disease severity was scored using the WHO 8-point ordinal scale for clinical improvement [25] with the following gradations: 3, hospitalized, no oxygen therapy; 4, oxygen by mask or nasal prongs; 5, non-invasive ventilation or high-flow oxygen; 6, intubation and mechanical ventilation; 7, ventilation with additional organ support. For analysis, we used three categories: 3, 4, and 5–7.

All participants were invited to outpatient follow-up visits at 3 and 12 months after hospital admission, including pulmonary function tests, blood sampling, and chest CT. Participants with chest CT findings consistent with COVID-19 sequelae after 3 months had a follow-up scan at 12 months. Fig 1 shows patient recruitment and attrition in the study. Ultimately, 124 patients completed a repeat CT scan at 12 months.

## Chest CT protocol, image review, and scoring

The participating centres used the same volumetric CT protocol across the various CT systems, and images were reconstructed with thin (0.9–1.25 mm) section thickness for evaluation. For detailed description of the protocol and image acquisition, see S1 File.

Three thoracic radiologists (T.M.A, C.M., J.R.R.) with at least 10 years of experience reviewed the CT scans from all centres in consensus and in a random order. The observers were blinded to the patient's clinical data. The readers registered the presence, extent, and

distribution of CT features using nomenclature recommended by the Fleischner Society [26], including GGO, parenchymal bands, reticular pattern, airspace consolidation, interlobular septal thickening, bronchiectasis and/or bronchiolectasis, and mosaicism.

The distribution and extent of disease was individually assessed in eight zones, four for each lung (above the aortic arch, between the aortic arch and the level of the carina, between the level of the carina and the level of the left inferior pulmonary vein, and below). The extent of affection in each zone was assigned a score based on the percentage of lung parenchyma with abnormality on a 0–3 scale (0, no involvement; 1, minimal; 2, moderate; 3, severe). An overall score of parenchymal involvement for each CT feature for each patient was derived by summing the scores of the four zones to an ordinal chest CT severity score (CSS) ranging from 0 (no involvement) to 12 (maximum involvement). The method was derived from a standardized method for evaluation of interstitial lung disease [27]. For detailed description of the review and scoring, see S1 File.

For regression analysis and easier visualization of changes in the four CT features of the CSS with highest prevalence of abnormalities (>20%), we recoded the CSS into an abbreviated CSS (aCSS) scale ranging from 0 to 3: 0 (none), CSS score of 0; 1 (mild), CSS 1–3; 2 (moderate), CSS 4–7; and 3 (severe), CSS 8–12.

## Statistical analysis

Descriptive statistics are presented using the mean (SD), median (25th–75th percentile), median (range), or number (%). Independent groups were compared using the t-test, chi-squared test, or Fisher's exact test, as appropriate. Paired comparisons of ordinal scores for CT abnormalities between 3 and 12 months after COVID-19 were analysed using the Wilcoxon signed rank test.

We analysed the aCSS for four CT features (GGO, parenchymal bands, mosaicism, and grade 1 reticular pattern, each scored on a 0–3 scale) using multivariable mixed effects ordinal regression analysis, with random effect for patient and fixed effects for other variables. Because there was only one participant with severe involvement (aCSS score 3) for parenchymal bands and reticulation, we combined the moderate/severe levels in these dimensions. As independent variables, we chose WHO severity rating (3, 4, 5–7), age (<60 vs ≥60 years), sex, and time of CT assessment (3 vs. 12 months). The number and choice of independent variables in the models were decided prior to the analysis, without statistical variable selection. We assessed whether there was a multiplicative interaction between WHO severity rating and time of CT assessment, but the interaction term was not statistically significant in any of the four models. The results are presented as odds ratios (OR) for being in a higher category of CT severity dimension score. We verified the proportional odds assumption using the Brant and Wolfe-Gould tests. The assumption was acceptable in models without random effects, as these functions cannot account for a mixed model.

We chose a significance level of 5%, using two-sided test. We used Stata SE version 17.0 (StataCorp. LLC, College Station, TX, USA) for analysis.

## Results

Participants with repeat CT scans (n = 124) were older than those with CT scans only at 3 months (n = 118), with mean age 62.3 (SD 13.3) and 53.8 (13.9) years, respectively (p<0.001). Those with two CT scans comprised a larger proportion of males, had lower education levels, more severe acute COVID-19, longer hospital stays, and increased need for ventilator treatment (Table 1). The 3-month CT scans were completed median (25th–75th percentile) 96 (84–

**Table 1. Demographics, comorbidity and variables during hospitalization for acute COVID-19.** Mean (SD) unless specified otherwise.

| | n | All Chest-CT at 3 months (n = 242) | n | Chest-CT at 3 and 12 months (n = 124) | n | Chest-CT only at 3 months (n = 118) | P* |
|---|---|---|---|---|---|---|---|
| Age (years) | 242 | 58.1 (14.2) | 124 | 62.3 (13.3) | 118 | 53.8 (13.9) | <0.001 |
| Sex (males), number (%) | 242 | 136 (56) | 124 | 78 (63) | 118 | 58 (49) | 0.031 |
| Norwegian origin, number (%) | 223 | 161 (72) | 112 | 85 (76) | 111 | 76 (68) | 0.23 |
| Education, years | 231 | | 118 | | 113 | | 0.017 |
| <11 | | 41 (18) | | 27 (23) | | 14 (12) | |
| 11–13 | | 64 (28) | | 37 (31) | | 27 (24) | |
| >13 | | 126 (54) | | 54 (46) | | 72 (64) | |
| Smoking status | 222 | | 112 | | 110 | | 0.74 |
| Never | | 129 (58) | | 62 (55) | | 67 (61) | |
| Previous smoker | | 87 (39) | | 47 (42) | | 40 (36) | |
| Current smoker | | 6 (3) | | 3 (3) | | 3 (3) | |
| *Comorbidity* | | | | | | | |
| Previous myocardial infarction | 241 | 15 (6) | 124 | 11 (9) | 117 | 4 (3) | 0.08 |
| Congestive heart failure | | | 124 | 4 (5) | 117 | 6 (5) | 0.53 |
| Diabetes | 241 | 21 (9) | 124 | 13 (10) | 117 | 8 (7) | 0.32 |
| Chronic obstructive pulmonary disease | 241 | 8 (3) | 124 | 4 (3) | 117 | 4 (3) | 1.00 |
| Charlson comorbidity index | 241 | | 124 | | 117 | | 0.69 |
| 0 | | 173 (72) | | 87 (70) | | 86 (74) | |
| 1 | | 41 (17) | | 21 (17) | | 20 (17) | |
| ≥2 | | 27 (11) | | 16 (13) | | 11 (9) | |
| *At admission* | | | | | | | |
| Systolic blood pressure (mmHg) | 238 | 134 (18) | 122 | 135 (20) | 116 | 132 (17) | 0.13 |
| Body temperature (˚C) | 235 | 37.6 (1.1) | 120 | 37.8 (1.1) | 115 | 37.5 (1.0) | 0.011 |
| Pulse rate (per min) | 236 | 85.1 (18.4) | 121 | 85.7 (16.1) | 115 | 84.6 (20.5) | 0.65 |
| Body mass index (kg/m2) | 184 | 27.7 (4.9) | 94 | 27.9 (5.6) | 90 | 27.4 (4.0) | 0.47 |
| *Laboratory tests at admission* | | | | | | | |
| B-Hemoglobin (g/dL) | 240 | 14.0 (1.5) | 123 | 14.0 (1.6) | 117 | 13.9 (1.4) | 0.76 |
| C-reactive protein (mg/L), median (25th–75th perc.) | 239 | 50 (19–112) | 123 | 88 (42–140) | 116 | 27 (9–67) | <0.001 |
| P-Creatinine (μmol/L) | 237 | 83.7 (47.1) | 123 | 88.0 (56.1) | 114 | 79.1 (34.5) | 0.15 |
| B-Leukocytes (x 10^9/L) | 240 | 8.5 (4.1) | 123 | 9.7 (4.4) | 117 | 7.3 (3.3) | <0.001 |
| *During hospitalization* | | | | | | | |
| Length of stay (days), median (25th–75th perc.) | 239 | 6 (3–12) | 124 | 8.5 (4.5–16) | 115 | 4 (2–7) | <0.001 |
| Ventilator treatment/intubation | 235 | 31 (13) | 120 | 25 (21) | 115 | 6 (5) | <0.001 |
| WHO severity rating | 241 | | 124 | | 117 | | <0.001 |
| 3 No oxygen treatment | | 84 (35) | | 27 (22) | | 57 (49) | |
| 4 Supplementary oxygen | | 116 (48) | | 66 (53) | | 50 (42) | |
| 5–7 High-flow oxygen/non-invasive or invasive ventilation/ECMO | | 41 (17) | | 31 (25) | | 10 (9) | |
| Positive PCR-test for SARS-CoV2 | 241 | | 124 | | 117 | | 0.52 |
| No | | 5 (2) | | 3 (2) | | 2 (2) | |
| Yes, before hospitalization | | 116 (48) | | 55 (44) | | 61 (52) | |
| Yes, during hospitalization | | 120 (50) | | 66 (53) | | 54 (46) | |

*t-test, chi-squared test or Fisher's exact test.

120) days (n = 241) post-admission, and the 12-month scans 398.5(364–461) days (n = 124) post-admission.

Among patients with Chest CT at 3 and 12 months (n = 126), 5 patients used steroids prior to hospitalization, and only an additional 2 patients were prescribed steroids during the hospital stay. Furthermore, 4 patients received interleukin-1 receptor antagonist (anakinra) and 1 received infusion with normal human IgG antibodies. In total, 8 of 126 patients were prescribed antibiotics for suspected secondary pulmonary or other infection. Information on oxygen use was not available at discharge. However, none of the patients used supplementary oxygen at the 12-month follow-up.

## Chest CT severity scores

Median CSS score (ranging from 0 to 12) for GGO decreased from 3 (25th–75th percentile: 0–12) at 3 months to 0.5 (0–12) at 12 months (p<0.001). A similar improvement was noted for mosaicism. Conversely, there was deterioration for parenchymal bands (p<0.001) (Table 2).

Transitions in individual CSS scores showed a similar trend (Fig 2).

In analysis of changes between 3 and 12 months in the four most prevalent CT abnormalities, stratified according to initial WHO disease severity, there was an improvement across all severity categories for GGO, and in the most severe categories (4 and 5–7) for mosaicism (Fig 3). For parenchymal bands, CSS scores increased between 3 and 12 months for those in the two least severe categories (3 and 4). Fig 4 illustrates the individual changes within each disease severity stratum.

## Abbreviated chest CT severity scores

The individual shifts in the aCSS scores (0–3 range) from 3 to 12 months were larger for GGO than for the other three CT features (Fig 5).

In multivariable mixed effects ordinal regression aCSS scores, for GGO, the odds of being in a more severe aCSS category at 12 months were 0.11 (95%CI 00.05 to 0.21, p<0.001) compared to 3 months after adjustment for age, sex and WHO severity (Table 3). The odds of being in a more severe aCSS category were 7.16 (1.18 to 43.47, p = 0.032) for WHO severity category 5–7 compared to category 3. The odds of being in a higher aCSS category were 4.80 (95%CI 1.33 to 17.6, p = 0.016) for age ≥60 compared to <60 years.

For parenchymal bands, there was a deterioration at 12 months compared to 3 months, OR 2.18 (95%CI 1.16 to 4,10, p = 0.015), and for WHO severity category 5–7 compared to category

**Table 2. Chest CT severity scores at 3 months and comparison between scores at 3 and 12 months in those with follow-up chest CT.** Median (range).

| Chest-CT severity score (0–12 scale) | All with Chest-CT at 3 months (n = 242) | Subset with follow-up Chest-CT twice (n = 124) | | |
|---|---|---|---|---|
| | | 3 months | 12 months | P* |
| Ground-glass opacities | 0 (0–12) | 3 (0–12) | 0.5 (0–12) | <0.001 |
| Parenchymal bands | 0 (0–7) | 0 (0–7) | 0.5 (0–8) | 0.003 |
| Mosaicism | 0 (0–8) | 0 (0–8) | 0 (0–7) | <0.001 |
| Reticular pattern grade 1 | 0 (0–8) | 0 (0–8) | 0 (0–6) | 0.13 |
| Reticular pattern grade 2 | 0 (0–12) | 0 (0–12) | 0 (0–7) | 0.09 |
| Reticular pattern grade 3 | 0 (0–4) | 0 (0–4) | 0 (0–0) | 0.16 |
| Bronchiectasis | 0 (0–8) | 0 (0–8) | 0 (0–8) | 0.094 |

*Wilcoxon sign rank test.

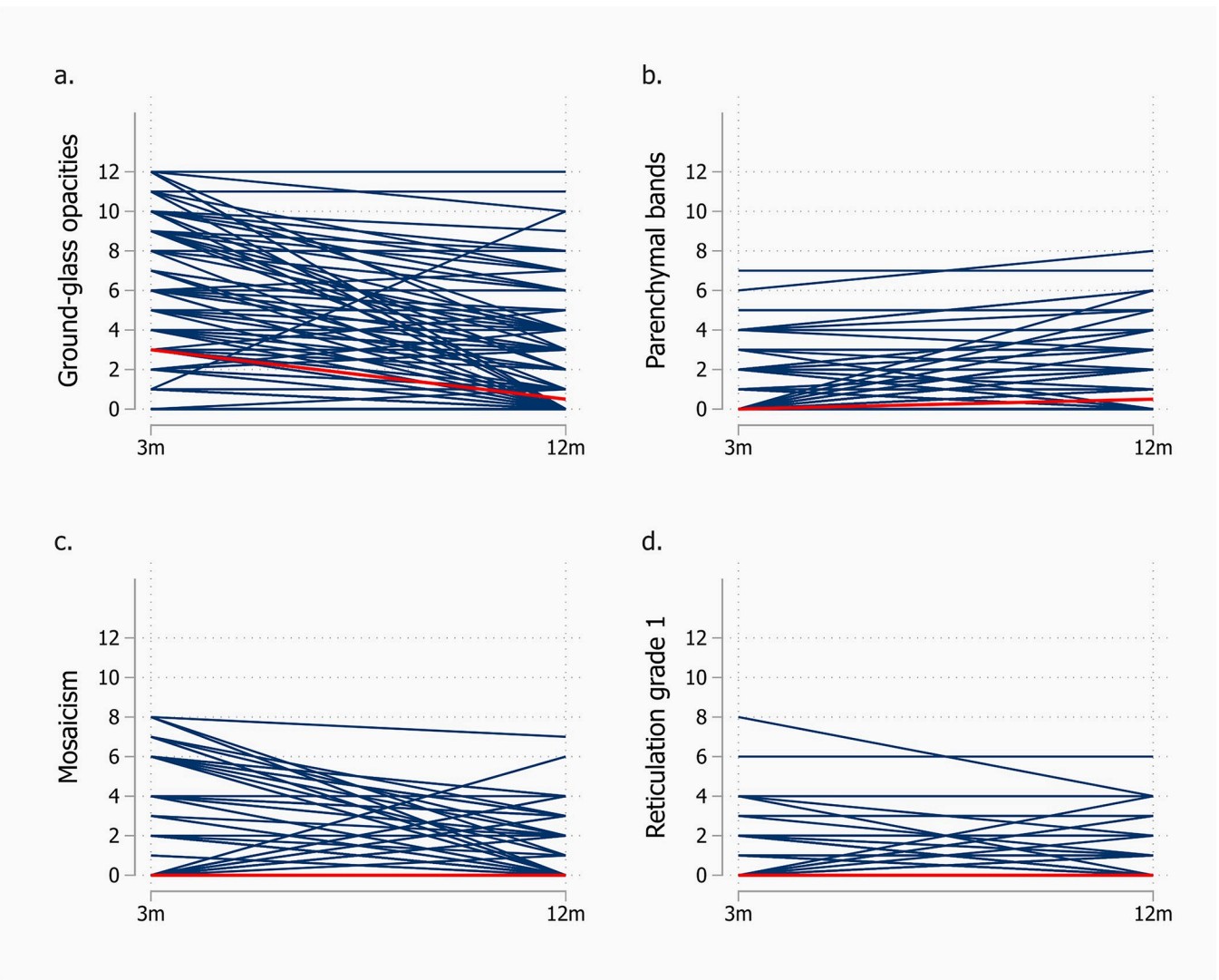

**Fig 2. Individual transfers of CT finding on ordinal 0–12 scale (12 most severe), from 3 (3m) to 12 months (12m) after hospital admission for COVID-19.** The lines are unweighted; hence, lines for several patients may be superimposed and do not reflect frequencies (n = 124). Red line connects median score at 3 and 12 months. The panels are: a. Ground-glass opacities, b. Parenchymal bands, c. Mosaicism, d. Reticular pattern grade 1.

3, OR 22.90 (95%CI 4.47 to 117.46, p<0.001). For mosaicism, there was an improvement at 12 compared to 3 months, with OR 0.33 (95%CI 0.16 to 0.66, p = 0.002).

## Discussion

In this study, GGO and mosaicism decreased markedly from 3 to 12 months as assessed by changes in semi-quantitative ordinal CT scores, while parenchymal bands increased. The decrease in GGO was consistent across all severity groups. Multivariable regression models of abbreviated ordinal scores, confirmed these changes over time. GGO were also associated with initial COVID-19 severity and age>60 years, and parenchymal bands were associated with initial disease severity.

### Ground-glass opacities

The finding of high CSS scores for lung GGO at 3 months with a decline in prevalence and CSS scores between 3 and 12 months after COVID-19 support previous reports using

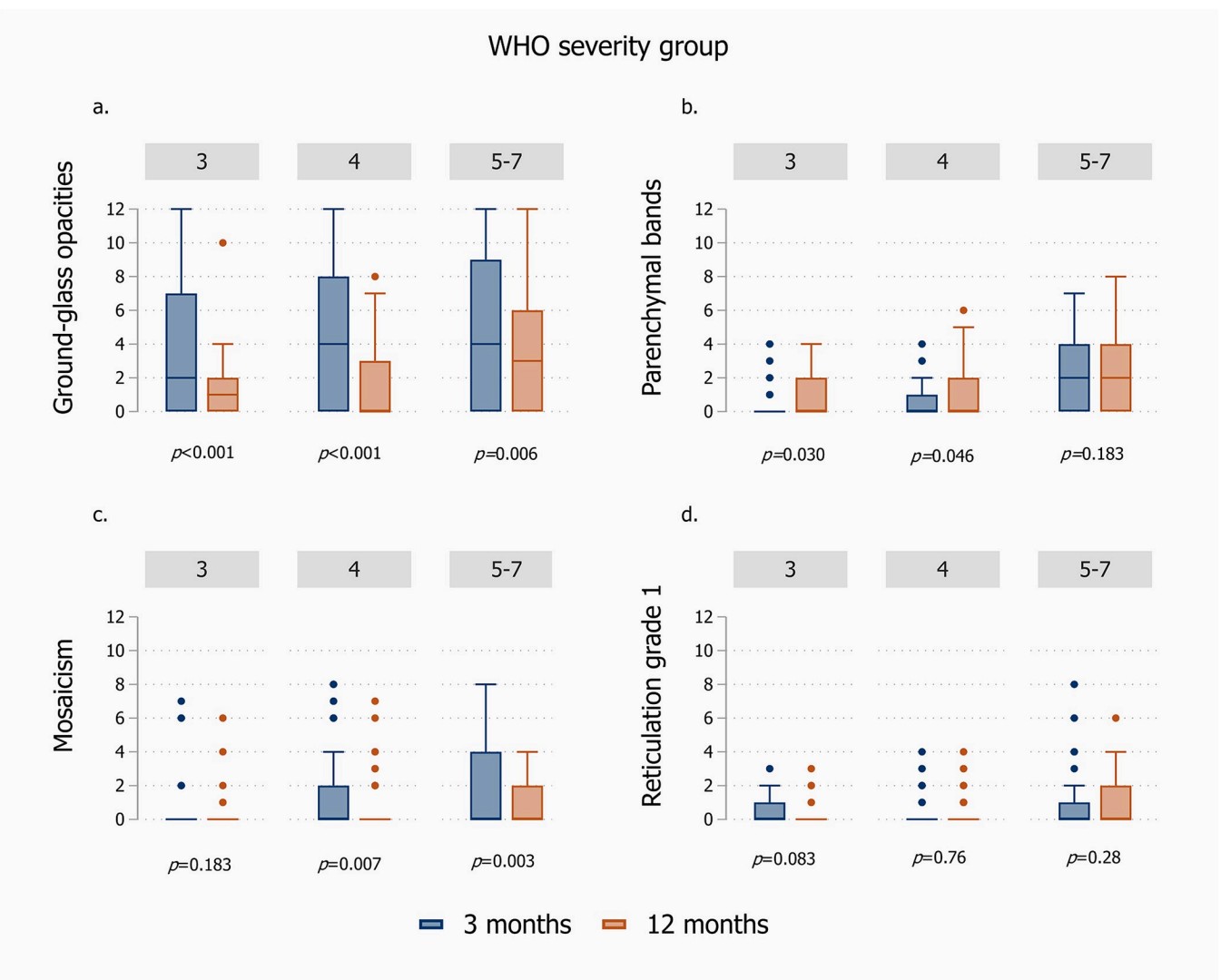

**Fig 3. Change in CT findings on ordinal Chest CT severity (CSS), from 0 (none) to 12 (most severe) from 3 to 12 months after hospital admission for COVID-19, stratified according to WHO disease severity during the hospital stay (3 = No oxygen treatment, 4 = Supplementary oxygen, 5–7 = High-flow oxygen/non-invasive or invasive ventilation/ECMO) (n = 124).** The panels are: a. Ground-glass opacities, b. Parenchymal bands, c. Mosaicism, d. Reticular pattern grade 1.

dichotomized data [11,12,28,29] or CT scores [30]. This was also supported in a multivariable model, after adjustment for relevant covariates. In a recent meta-analysis, the pooled prevalence of GGO after 1 year was 23.8% (range 2.4–67.7%) with substantial heterogeneity between studies [8], reflecting variations in participant selection, severities, and study designs. The gradual resolution over time suggests that GGO is compatible with non-specific interstitial pneumonia patterns or post-organizing pneumonia that may resolve over time [8,31].

## Fibrotic-like changes

In our study, grade1 reticular pattern and parenchymal bands were relatively common at 3 months, in line with findings in a meta-analysis with median follow-up of 3 months [22]. Early during the pandemic, parenchymal bands were a common finding after COVID-19 [32]. In some studies, such findings were not explicitly specified, but may have been included in a

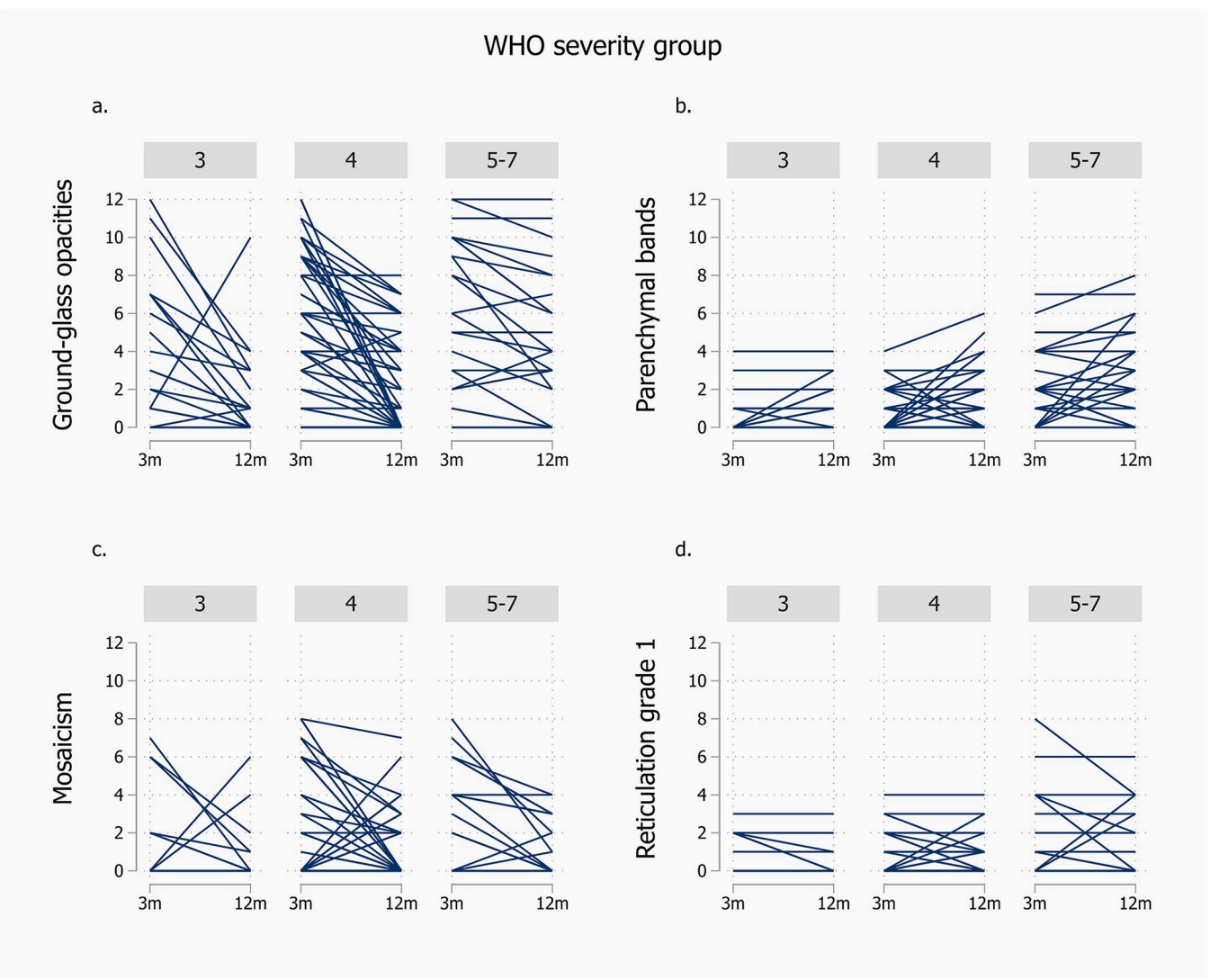

**Fig 4. Individual transfers of CT finding on ordinal 0–12 scale (12 most severe), from 3 (3m) to 12 months (12m) after hospital admission for COVID-19, stratified according to WHO disease severity during the hospital stay (3 = No oxygen treatment, 4 = Supplementary oxygen, 5–7 = High-flow oxygen/non-invasive or invasive ventilation/ECMO).** The lines are unweighted; hence, lines for several patients may be superimposed and do not reflect frequencies (n = 124). The panels are: a. Ground-glass opacities, b. Parenchymal bands, c. Mosaicism, d. Reticular pattern grade 1.

broader category termed "fibrotic-like abnormalities" [10,16,30,33], which varies across studies and may have led to overestimation of the prevalence [7,33]. Parenchymal bands in thin-section chest CT may reflect pleuroparenchymal fibrosis. However, they often occur after asbestos exposure and are not commonly related to UIP (usual interstitial pneumonia) or other idiopathic interstitial lung diseases [26,34].

In the present study, the severity of grade 1 reticular pattern remained unchanged from 3 to 12 months, while the severity scores for parenchymal bands increased. These findings are compatible with some previous reports, e.g., a systematic review with 1-year follow-up, including little change between 4–7 months and 1-year follow-up [7]. A recent study reported fibrotic-like changes in 19% (32/169) at 4 months and persistence in 95% (18/19) after 16 months, but the number with longitudinal follow-up was small [9].

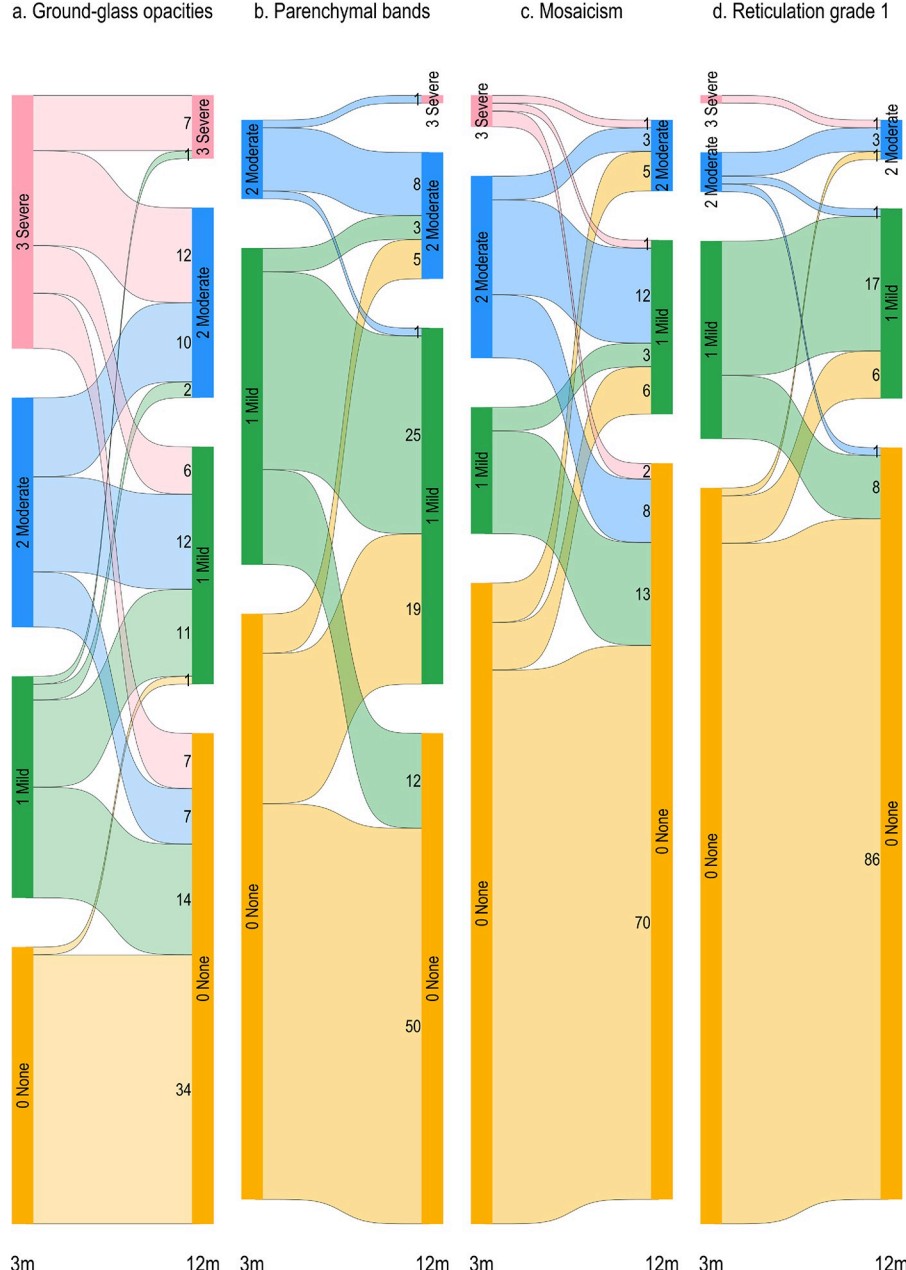

**Fig 5. Change in CT scores on abbreviated Chest CT severity (aCSS) scale from 3 (3m) to 12 months (12m).** Scale categories are: 0 (none), 1 (mild), 2 (moderate), 3 (severe). Widths of flows represent frequencies. The panels are: a. Ground-glass opacities, b. Parenchymal bands, c. Mosaicism, d. Reticular pattern grade 1.

CSS scores for parenchymal bands deteriorated for participants in the two least severe WHO categories (3 and 4), while no change was observed in the most severe category (5–7). Deterioration in parenchymal bands over time raises concerns about the development of chronic lung conditions post-COVID-19, also in less severe cases. However, our findings also indicate that secondary pulmonary fibrosis was more common in patients with severe COVID-19.

**Table 3. Odd ratios for being in a higher category of abbreviated chest CT severity dimension score (0–3 scale).** Multivariable mixed effects ordinal regression analysis of radiological scores. (n = 248 CT scans; 124 patients).

| Covariate | n | Ground-glass opacities | | | Parenchymal bands | | | Mosaicism | | | Reticular pattern grade1 | | |
|---|---|---|---|---|---|---|---|---|---|---|---|---|---|
| | | Odds ratio | 95% Conf. Interval | P | Odds ratio | 95% Conf. Interval | P | Odds ratio | 95% Conf. Interval | P | Odds ratio | 95% Conf. Interval | P |
| Who severity rating, grouped | | | | | | | | | | | | | |
| 3* No oxygen | 54 | 1 | | | 1 | | | 1 | | | 1 | | |
| 4 Oxygen | 132 | 1.30 | (0.28 to 6.01) | 0.73 | 1.2 | (0.32 to 4.51) | 0.79 | 2.57 | (0.71 to 9.27) | 0.15 | 0.65 | (0.08 to 5.20) | 0.68 |
| 5–7 High-flow /NIV/ Ventilator/ECMO | 62 | 7.16 | (1.18 to 43.47) | 0.032 | 22.9 | (4.47 to 117.46) | <0.001 | 3.69 | (0.90 to 15.17) | 0.071 | 6.42 | (0.57 to 71.90) | 0.13 |
| Age | | | | | | | | | | | | | |
| <60 years | 116 | 1 | | | 1 | | | 1 | | | 1 | | |
| > = 60 years | 132 | 4.80 | (1.33 to 17.26) | 0.016 | 1.98 | (0.69 to 5.71) | 0.21 | 0.36 | (0.14 to 0.97) | 0.043 | 3.86 | (0.67 to 22.40) | 0.13 |
| Sex | | | | | | | | | | | | | |
| Female* | 92 | 1 | | | 1 | | | 1 | | | 1 | | |
| Male | 156 | 0.48 | (0.13 to 1.73) | 0.26 | 0.94 | (0.32 to 2.76) | 0.91 | 1.81 | (0.65 to 5.01) | 0.26 | 6.65 | (1.02 to 43.43) | 0.048 |
| Months after admission | | | | | | | | | | | | | |
| 3* | 124 | 1 | | | 1 | | | 1 | | | 1 | | |
| 12 | 124 | 0.11 | (0.05 to 0.21) | <0.001 | 2.18 | (1.16 to 4.10) | 0.015 | 0.33 | (0.16 to 0.66) | 0.002 | 0.74 | (0.30 to 1.81) | 0.51 |
| *Referent | | | | | | | | | | | | | |
| ICC subject | | 0.712 (0.573 to 0.820) | | | 0.592 (0.408 to 0.754) | | | 0.474 (0.268 to 0.690) | | | 0.882 (0.703 to 0.960) | | |
| AIC | | 580.1 | | | 413.8 | | | 413.5 | | | 280.7 | | |

ICC = intraclass correlation coefficient; AIC = Akaike's information criterion; NIV = non-invasive ventilation; ECMO = extra-corporeal membrane oxygenation.

## Honeycombing/Traction bronciectasis

In our study, grade 3 macrocystic reticular pattern, corresponding to honeycombing, was negligible and in agreement with previous reports at 12 months after COVID-19 infection [10,15,35]. Therefore, the risk of manifest UIP-like fibrosis after post COVID-19 seems small.

We noted bronchial dilatation (traction bronchiectasis or traction bronchiolectasis) in only a few patients, with minimal observable changes between 3 and 12 months. This is in line with a low prevalence reported in other studies [8,13,28]. Bronchial dilatation may suggest fibrosis, but the relevance of the finding is unclear as it may be reversible following COVID-19 [36,37].

It is notable that pre-existing interstitial lung abnormalities, including honeycombing and bronchial dilatation, may be present in up to 9.7% of cases [38–40], potentially predating the onset of COVID-19 infection. Therefore, an important limitation with follow-up studies after COVID-19 is the lack of CT studies prior to the infection for comparison.

## Mosaicism

Mosaicism on inspiratory scans [26] were relatively common at 3 months and declined between 3 and 12 months. We did not systematically use paired inspiratory and expiratory scans, but nine patients with mosaicism on inspiratory scans, had air trapping consistent with small airways disease on supplementary expiratory scans. Small airways disease is common in airways infections including COVID-19 [41,42], but air trapping is common also in healthy subjects without suspected small airways disease [43].

## Association of CT abnormalities with disease severity

In multivariable analysis of aCSS, we have shown that the extent of GGO was associated with initial COVID-19 severity and age>60 years. Furthermore, parenchymal bands were associated with initial disease severity. Some studies have investigated associations with initial disease severity, with disparity between findings, and most studies are not adjusted for covariates. At about 1 year after COVID, no association with initial severity was shown for any chest CT feature [13], while another study reported an association for a global CT score and GGO, but not for other CT features at 12 months [16]. In multivariable analysis, increasing age was associated with an abnormal CT score [16], as in the present study, but disease severity was not examined. CT abnormality at 12 months were also associated with oxygen supply, and hospital length of stay, as markers of severity [28].

A recent meta-analysis did not reveal any difference in residual CT abnormalities at 1 year between mild/moderate and severe patients [7]. In a meta-regression analysis of 14 studies, the percentage of people with severe or critical COVID-19 or relevant subject characteristics, such as age, sex, smoking status, hypertension or diabetes, were not associated with the 1-year prevalence of any residual lung abnormalities, i.e., an aggregate variable at study level [8].

We have previously shown that there was an improvement in total lung capacity and diffusion capacity of the lungs for carbon monoxide ($D_{LCO}$) from 3 to 12 month after hospitalization for COVID-19, and that there was little association of any fibrotic-like finding on chest-CT with $D_{LCO}$ or modified Medical Research Council dyspnea scores after 12 months in multivariable analysis [17]. Therefore, we have not repeated similar analyses with ordinal radiology scores.

## Strengths and limitations

The subjects in this study represent about 30% of surviving patients with COVID-19 in Norwegian hospitals during the recruitment period in 2020 (882 admitted with COVID-19 as main diagnosis, and 91 COVID-19-associated death in hospitals) [44]. The attrition rate in the study was low. Therefore, the study should be reasonably representative of survivors after hospitalization for COVID-19 during the first wave in Norway. The proportion of patients admitted to the ICU was lower than in some other reports, and the mortality and complication rate was low, probably reflecting that Norwegian hospitals did not have a severely constrained hospital capacity and admitted many less severely ill patients than in some other countries. The CT images were reviewed at one hospital, with a limited number of readers in consensus, which should ease standardization. The protocol included both prone and supine acquisitions, which enabled us to avoid misinterpreting random gravity-dependent atelectasis as GGO or reticular opacities. We used multilevel ordinal regression models that are suitable for such data [20], while it is not unusual that the analysis of such ordinal data is oversimplistic or inappropriate [21].

The sample size was larger than many other longitudinal studies of CT after COVID-19, but still small. This limited the number of independent variables in multivariable analysis. Paired inspiratory and expiratory scans were not included in the CT protocol. All patients were hospitalized during the first wave of COVID-19 in 2020, when the genetic line B1 (Pangolin nomenclature) dominated in Norway with the exception of February 2020 [45]. Therefore, our sample might not be representative for the extent and severity of chest CT abnormalities for later corona virus variants, such as omicron, which lead to less extensive disease and a better prognosis [46].

## Implications for clinical practice

Understanding the long-term pulmonary consequences of COVID-19 is important for guiding clinical management and informing public health strategies. Consequently, longitudinal

follow-up studies are essential to confirm or rule out irreversible lung damage following COVID-19. This study has also confirmed that the time to resolution for GGO is shorter than for some fibrotic-like abnormalities, as previously noted [7,8,22]. We have also identified disease severity and age as risk factors or prognostic indicators, suggesting that a closer eye on patients with severe disease and age >60 might be sensible in follow-up after COVID-19. However, it is possible that many of the radiological abnormalities were detected in asymptomatic patients, but we have not presented data on this here.

## Conclusion

This study has shown high CSS for GGO at 3 months, but also parenchymal bands and mosaicism were common. Between 3 and 12 months, GGO and mosaicism decreased, while the presence of parenchymal bands increased, which were shown clearly by changes in the ordinal CSS and aCSS in multivariable models. Finally, those with severe COVID-19 had more persistent GGO and parenchymal bands.

## Supporting information

**S1 File. Details on chest CT protocol, and review and scoring of CT images.**
(DOCX)

## Author Contributions

**Conceptualization:** Trond Mogens Aaløkken, Haseem Ashraf, Gunnar Einvik, Ole Henning Skjønsberg, Knut Stavem.

**Data curation:** Trond Mogens Aaløkken, Gunnar Einvik, Tøri Vigeland Lerum, Carin Meltzer, Jezabel Rivero Rodriguez, Knut Stavem.

**Formal analysis:** Knut Stavem.

**Funding acquisition:** Haseem Ashraf.

**Investigation:** Trond Mogens Aaløkken, Haseem Ashraf, Gunnar Einvik, Tøri Vigeland Lerum, Carin Meltzer, Jezabel Rivero Rodriguez, Knut Stavem.

**Methodology:** Trond Mogens Aaløkken, Haseem Ashraf, Gunnar Einvik, Carin Meltzer, Jezabel Rivero Rodriguez, Ole Henning Skjønsberg, Knut Stavem.

**Project administration:** Gunnar Einvik, Tøri Vigeland Lerum, Ole Henning Skjønsberg, Knut Stavem.

**Resources:** Haseem Ashraf, Gunnar Einvik, Tøri Vigeland Lerum.

**Software:** Knut Stavem.

**Supervision:** Trond Mogens Aaløkken, Haseem Ashraf, Ole Henning Skjønsberg, Knut Stavem.

**Validation:** Trond Mogens Aaløkken, Tøri Vigeland Lerum, Knut Stavem.

**Visualization:** Knut Stavem.

**Writing – original draft:** Trond Mogens Aaløkken, Knut Stavem.

**Writing – review & editing:** Trond Mogens Aaløkken, Haseem Ashraf, Gunnar Einvik, Tøri Vigeland Lerum, Carin Meltzer, Jezabel Rivero Rodriguez, Ole Henning Skjønsberg, Knut Stavem.

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
