## [Decision Letter · Decision Letter 0]

29 Jan 2024

PONE-D-23-42962CT abnormalities 3 and 12 months after hospitalization for COVID-19 and association with disease severity: a prospective cohort studyPLOS ONE

Dear Dr. Stavem,

Thank you for submitting your manuscript to PLOS ONE. After careful consideration, we feel that it has merit but does not fully meet PLOS ONE’s publication criteria as it currently stands. Therefore, we invite you to submit a revised version of the manuscript that addresses the points raised during the review process.

We look forward to receiving your revised manuscript.

Kind regards,

Julie Zhu

Academic Editor

PLOS ONE

Journal Requirements:

"Haseem Ashraf reports grant from Boehringer-Ingelheim, Norway. Knut Stavem reports personal fees from UCB Pharma and MSD Norway, unrelated to this study."

We note that you received funding from a commercial source: UCB Pharma and MSD Norway

Within this Competing Interests Statement, please confirm that this does not alter your adherence to all PLOS ONE policies on sharing data and materials by including the following statement: ""This does not alter our adherence to PLOS ONE policies on sharing data and materials.” (as detailed online in our guide for authors http://journals.plos.org/plosone/s/competing-interests).  If there are restrictions on sharing of data and/or materials, please state these. Please note that we cannot proceed with consideration of your article until this information has been declared. 

3. We noted in your submission details that a portion of your manuscript may have been presented or published elsewhere:

"Yes. The cohort has been presented before, including follow up at 3 and 12 months. 

Crude prevalence of radiology findings has been presented along with lung function/gas diffusion findings, but these prevalences are not presented/repeated in the present  publication. The paper focuses on ordinal scores of CT images, including comprehensive analysis of these scores."

**Additonal Editor Comments:**

Thank you for submitting this important manuscript. Would appreciate your review, revision and resubmission according to reviewers' comments.

Reviewers' comments:

Reviewer's Responses to Questions

**Comments to the Author**

1. Is the manuscript technically sound, and do the data support the conclusions?

Reviewer #1: Yes

2. Has the statistical analysis been performed appropriately and rigorously? 

Reviewer #1: Yes

3. Have the authors made all data underlying the findings in their manuscript fully available?

Reviewer #1: No

4. Is the manuscript presented in an intelligible fashion and written in standard English?

Reviewer #1: Yes

5. Review Comments to the Author

Reviewer #1: Referees comments on CT abnormalities 3 and 12 months after hospitalization for COVID-19 and association with disease severity: a prospective cohort study

PONE-D-23-42962

Studies assessing the long term effects of Covid-19 is of huge importance. The virus was associated to intense changes in lung function, and CT-imaging did during the acute phase showed profound pathology. The resolution of CT-changes is of interest but must of course be put in context of the physiological associated effects. Lung function oxygenation, and capacity to regain pre-infectious working capacity is of outmost importance.

The present study from Norway presents results from 3 and 12 months CT follow-up of a huge groups of Covid-19 patients. It shows that ground-glass opacities (GGO) and mosaicism decreased, while parenchymal bands increased from 3 to 12 months. Persistent GGO were associated with initial COVID-19 severity and age ≥60 years. It provides however no information around lung function; patients assessment of breathing and work capacity or physiological measures of the lung function.

The CT-findings and analysis in relation to patients’ demographics, comorbidities, on acute phase disease severity is done in a statistically sophisticated and reasonable fashion. The main weakness is the lack of information around the association to lung function and even more important as well as patients assessed outcome.

Some questions that I see could be of value to clarify;

How many of the patients were provided steroids, and dose of steroids if given.

How many of patients experienced any kind further lung infection during the follow-up period, bronchitis, pneumonia, re-infection with Covid or influenza

Where any of the patients included treated with inhaled steroids or other inhaled medications during the follow-up period

Where any of the patients on systemic steroids e.g. for arthritis

How many of the patients were on oxygen therapy after discharge and for how long time

6. PLOS authors have the option to publish the peer review history of their article (what does this mean?). If published, this will include your full peer review and any attached files.

Reviewer #1: No

---

## [Author Response · Author response to Decision Letter 0]

21 Mar 2024

See enclosed cover letter and rebuttal letter

---

## [Decision Letter · Decision Letter 1]

16 Apr 2024

CT abnormalities 3 and 12 months after hospitalization for COVID-19 and association with disease severity: a prospective cohort study

PONE-D-23-42962R1

Dear Dr. Stavem

We’re pleased to inform you that your manuscript has been judged scientifically suitable for publication and will be formally accepted for publication once it meets all outstanding technical requirements.

Kind regards,

Julie Zhu

Academic Editor

PLOS ONE

---

## [Editor Report · Acceptance letter]

26 Apr 2024

PONE-D-23-42962R1 

PLOS ONE

Dear Dr. Stavem, 

I'm pleased to inform you that your manuscript has been deemed suitable for publication in PLOS ONE. Congratulations! Your manuscript is now being handed over to our production team.

Kind regards, 

on behalf of

Dr. Julie Zhu 

Academic Editor

PLOS ONE